# Reproduction and Extension of "Queens are Powerful too: Mitigating Gender Bias in Dialogue Generation"

## Reproducibility Summary

**Scope of Reproducibility**

The main claims we are trying to reproduce are that bias controlled training or combining counterfactual data augmentation, the positively biased data collected by Dinan et al. [5], and bias controlled training for the LIGHT dataset yields generated dialogue in which the percent of gendered words and male bias closely match the ground truth.

**Methodology**

We fine-tuned a transformer model, pre-trained on Reddit data [1], using the ParlAI API [8] with counterfactual data augmentation, positively biased data collection, bias controlled training, and all three bias mitigation techniques combined, as discussed in the original paper [5]. We implemented counterfactual data augmentation and bias controlled training ourselves. All models were trained and evaluated using a single NVIDIA Tesla P100 PCIe GPU, which took between 1.3 and 4.6 GPU hours approximately.

**Results**

Overall, our results support the main claims of the original paper [5]. Although the percent gendered words and male bias in our results are not exactly the same as those in the original paper [5], the main trends are the same. The main difference is lower male bias for the baseline model in our results. However, our findings and the trend similarities between our results and those obtained by Dinan et al. [5] demonstrate that bias controlled training or combining all three bias mitigation techniques can effectively control the amount of gender bias present in the model generated responses, supporting Dinan et al.'s claims [5].

**What was easy**

When reproducing the original paper [5], implementing counterfactual data augmentation and bias controlled training was easy since these techniques were well-described in the original paper [5]. Also, combining all three bias mitigation techniques was simple, as we applied the same techniques used to implement each bias mitigation method individually.

**What was difficult**

The only difficulty we encountered, albeit minor, was learning how to use ParlAI, which was necessary to use the same model as in the original paper [5]. However, after reading through the ParlAI documentation and experimenting with the ParlAI Google Colaboratory tutorial [10], we understood how to use ParlAI to fine-tune the model, pre-trained on Reddit conversations [1], for the datasets we create.

**Communication with original authors**

We communicated with Emily Dinan, an author of the original paper [5], who clarified what model was used in the original paper [5] and provided us with the command to download the model as well as the hyperparameter settings used when fine-tuning.

# 1   Introduction

Ad-hoc methods for mitigating social bias in natural language data remain an active area of modern research. As transfer learning with pre-trained models such as BERT [3] and GPT-2 [9] continue to be pervasive, the inherent issues in their training data have come to light. Large corpora of unstructured text from the Internet reflect the biases and inequalities of society, and are consequently learned by these models and their fine-tuned variants. To this end, Dinan et al. [5] proposed three techniques to specifically mitigate gender bias in fine-tuned language models, using the LIGHT dataset [11] as an example. The LIGHT dataset is a crowdsourced collection of dialogues spoken between "personas," characters played by either humans or models, in a fantasy adventure game, LIGHT [11]. Dinan et al. applied the following techniques to this dataset: 1) counterfactual data augmentation, in which gendered words are replaced with their opposite, i.e., replacing "he" with "she"; 2) positively biased data collection, in which new, less biased female character personas and dialogues are created via crowd-sourcing; and 3) bias controlled training, in which the dialogue is placed in groups based on the number of gendered words it contains and this group number is included with the dialogue as a special token when training the model [5]. The model itself is a transformer pre-trained on a dataset of Reddit conversations [1] and then fine-tuned on LIGHT using the three techniques described above, individually, as well as one combining all three techniques.

# 2   Scope of reproducibility

The aim of this paper is to evaluate the following hypotheses made by Dinan et al. [5] by reproducing their experiments.

- Combining counterfactual data augmentation, the positively biased data collected by Dinan et al. [5], and bias controlled training for the LIGHT dataset yields generated dialogue in which the percent of gendered words and male bias closely match the ground truth.

- Bias controlled training for the LIGHT dataset yields generated dialogue in which the percent of gendered words and male bias closely match the ground truth.

# 3   Methodology

We fine-tuned the transformer model, pre-trained on Reddit data [1], using the ParlAI API [8] with counterfactual data augmentation, positively biased data collection, bias controlled training, and all three bias mitigation techniques combined, as discussed in the original paper [5]. We generated training, test, and validation datasets for counterfactual data augmentation and bias controlled training from the original LIGHT dialogue dataset. We also formatted the dataset used for each bias mitigation technique, extracting the dialogue from each dataset and placing it in the proper format, such that everything said in the dialogue so far is used to predict the next response in the dialogue, which is the label. All models were trained and evaluated using a single NVIDIA Tesla P100 PCIe GPU.

## 3.1   Model descriptions

Dinan et al. [5] used a transformer with 8 encoder layers, 8 decoder layers, embedding dimension of 512, and 16 attention heads. This model was pre-trained on Reddit conversations from the pushshift.io Reddit dataset, which contains 2.2 billion samples for training after removing comments that contain URLs or that are less than 5 characters long [5]. Specifically, the model was trained on all comments in each thread and learned to predict the next comment in the thread [5]. Thus, this pre-training makes the model well-suited for the dialogue generation task [1]. The model contains $87,508,992$ trainable parameters and the training objective is to minimize the cross entropy loss on the original and augmented LIGHT dialogues.

## 3.2   Datasets

We used the ParlAI API command from the paper's ParlAI project page [4] to obtain the following data: the LIGHT dataset [11], a list of counterfactuals, a list of gendered words [12], and the positively biased data collected by Dinan et al. [5]. The LIGHT dataset and positively biased data collected by Dinan et al. contain information about interactions between characters in the game, LIGHT, such as the character names and personas, dialogue, and environment where the interaction took place, to name a few. The LIGHT dataset contains approximately $11,000$ interactions and $111,000$ utterances [11]. An utterance is a single occurrence of a character talking during a dialogue. The LIGHT dataset is used to fine-tune the baseline model.

Each bias mitigation method employed by Dinan et al. [5] also requires fine-tuning the pre-trained model on a new dataset. For counterfactual data augmentation, we used the list of counterfactuals to replace every gendered word, according to the list of gendered words from Zhao et al. [12], in the LIGHT dialogue dataset with its counterfactual. The list of gendered words [12] has $1,049$ words. The list of counterfactuals contains each gendered word and its opposite gendered counterpart. For example, the counterfactual for "he" is "she". In addition, the list of counterfactuals, containing $421$ words, was constructed by Dinan et al. [5] using the list of gendered words from Zhao et al. [12].

For positively biased data collection, Dinan et al. crowdsource new dialogue data, asking workers to create dialogue assuming gender equality [5]. This dataset contains $507$ interactions and $6,658$ utterances. Given the time and resource constraints, we used Dinan et al.'s positively biased data [5] rather than crowdsourcing the data ourselves.

For bias controlled training, we appended "fx my" after the last utterance in an episode, which is a portion of a dialogue between two characters, based on the label, which is the next utterance in the dialogue. In "fx my," x is 1 if there is at least one female gendered word in the label and 0 otherwise, and y is 1 if there is at least one male gendered word in the label and 0 otherwise. Thus, each label falls into one of four bins: "f0 m0" which has no gendered words; "f0 m1" which has no female gendered words but at least one male gendered word; "f1 m0" which has at least one female gendered word but no male gendered words; and "f1 m1" which has at least one female and one male gendered word. Placing the dialogue labels in these bins causes the model to learn the gender bias present in an utterance, allowing us to specify the desired gender bias in the model's generated dialogue using one of the four bins. We used the list of gendered words from Zhao et al. [12] to determine the number of gendered words and proper bin for each label and model generated utterance.

We split the datasets used for fine-tuning each model into approximately $90\%$ for training and $10\%$ for an unseen test set. The training set was further split into $80\%$ for training and $20\%$ for validation.

## 3.3 Hyperparameters

As previously mentioned, the model, pre-trained on Reddit conversations, has $8$ encoder layers, $8$ decoder layers, $16$ attention heads, and an embedding dimension of $512$ [1]. In addition, this model has $2,048$ nodes in the hidden layer, uses GeLU activation function, and truncates each dialogue to at most $512$ characters and each label to at most $128$ characters. Other hyperparameters for each model are an initial learning rate of $3.1e - 7$, memory-efficient Adam optimizer, gradient clipping of $0.1$, inverse square root learning rate scheduler with a decay factor of $0.5$ and patience of $3$, no activation or attention dropout, batch size of $20$, and dropout of $0.1$ or $0.15$ depending on hyperparameter tuning results. Emily Dinan, one of the authors of the original paper [5], provided some of the hyperparameter values, but we reduced the batch size due to memory constraints with Google Colaboratory resources. Since most hyperparameters were provided by Emily Dinan and the learning rate is adjusted by the inverse square root learning rate scheduler and batch size could not be increased due to GPU limitations, the only remaining hyperparameter that we could effectively tune to improve perplexity, based on our experience with deep NLP models, particularly pre-trained transformers, was dropout. Thus, we tuned dropout, applied to the embeddings and before layer normalization, for the model combining all three bias mitigation techniques, since this model provided the best results according to the original paper [5], to obtain lower perplexity on the validation set. In order to tune dropout, we increased dropout in increments of $0.025$, starting from a value of $0.1$, which was given by Emily Dinan, up to $0.2$. After training a number of models with different dropouts, we found that $0.15$ dropout resulted in the lowest perplexity. In addition, for the extension with neutral, generated data, we again tuned dropout, and found $0.15$ to be the optimal value.

## 3.4 Experimental setup and code

Similar to the Reddit dataset used for pre-training the model as well as the training done by Dinan et al. [5], we generated the datasets based on the entire history of conversations so far, predicting the next utterance in each conversation. For each bias mitigation technique and combining all three techniques, we generated the datasets from the original conversations in the LIGHT dataset [11] for training, evaluation, and response generation. Using ParlAI's API, we fine-tuned 5 versions of the model, pre-trained on Reddit conversations [1]: baseline, counterfactual data augmentation, positively biased data collection, bias controlled training, and all three bias mitigation techniques combined. When fine-tuning each model, the best model is saved according to the perplexity on the validation set. As long as the perplexity on the validation set continues to improve, the model continues training and at every quarter epoch, the version of the model achieving the lowest perplexity on the validation set is saved. If the model does not improve after 10 quarter epochs, training will be automatically stopped to avoid overfitting or unnecessary training. After training is complete, we run further evaluation to obtain F1 scores on the validation and test datasets as well as F1 scores pertaining to the labels for each bin for these two datasets. Finally, we pass every dialogue episode in the test set through the

model to generate responses. These generated responses are used to compute statistics defined by Dinan et al. [5] to evaluate gender bias in generated responses from the model.[1]

All experiments were run on Google Colaboratory using a single NVIDIA Tesla P100 PCIe GPU. After fine-tuning each model, the labels in the test set are split into the bias controlled training bins and within these bins, each model's generated utterances are also grouped into the same bins. This allowed us to compute the percent gendered words and male bias for the generated utterances within each bin of labels for the test set. In addition, we computed the F1 score for predicted tokens in generated responses separately for each bin of test labels.

### 3.5 Computational requirements

The model used by Dinan et al. in the original paper [5] was pre-trained on Reddit conversations in the same manner as the polyencoder transformer model from Humeau et al. [7], and contains the same number of encoder layers, decoder layers, attention heads, and embedding dimension size. Training the polyencoder transformer on the ConvAI2 dataset, which has about $131,000$ elements [6], took 2.7 hours using 8 NVIDIA Volta 100 GPUs [7]. Since the polyencoder transformer has about $20\%$ more parameters than the model used by Dinan et al. and the LIGHT dataset is about $15\%$ smaller than the ConvAI2 dataset, we estimated it took Dinan et al. about 2.3 hours or less, which is $85\%$ of 2.7 hours, using 8 GPUs to fine-tune each model or about 11.5 hours total for all 5 models.

We initially estimated we could also fine-tune all 5 models in approximately 11.5 hours using Google Cloud Platform. Instead, we used a single NVIDIA Tesla P100 PCIe GPU on Google Colaboratory. During training, each model required about 16 GB of GPU memory, maximizing the GPU memory available with the aforementioned batch size of 20. Table 1 lists runtime information for fine-tuning each model, where the model combining all three bias mitigation techniques uses dropout of $0.15$ for the embeddings and before layer normalization, as previously mentioned. The runtime for this model with other values for dropout was approximately the same. The actual training time for our models was substantially lower than our estimate, likely due, at least in part, to the unpredictability of Google Colaboratory providing the full computational GPU resources assigned to a particular session.

| Model | Number of Epochs | Training Time (GPU Hours) | Average Runtime per Epoch (GPU Hours) |
|---|---|---|---|
| Baseline | 7.51 | 1.32 | 0.18 |
| Counterfactual Data Augmentation | 4.75 | 1.63 | 0.34 |
| Positively Biased Data Collection | 7.26 | 1.40 | 0.19 |
| Bias Controlled Training | 7.76 | 1.38 | 0.18 |
| All 3 Bias Mitigation Techniques | 6.58 | 4.63 | 0.70 |

Table 1: Computational Requirements for Training each Model

## 4 Results

Below are the results from reproducing and extending the experiments in the original paper [5]. Overall, our results support the hypotheses previously identified. Further discussion of the results in relation to the hypotheses is provided below. We also implement 3 extensions to the original paper [5], two of which are aimed at addressing the high time and monetary cost of positively biased data collection, which requires crowdsourcing data.

Figure 1 shows the percent gendered words, percent male bias, and F1 score of each model's generated utterances for conversations in the test set, separated according to the test label bins, where "Baseline" is the model trained only on the LIGHT dataset, "CDA" is counterfactual data augmentation, "Pos Data" is positively biased data collection, "Bias" is bias controlled training, and "All" combines all three bias mitigation techniques. In Figure 1, each set of three graphs corresponds to one of the four bias controlled training bins for test labels. The results shown in Figure 1 are quite similar to those in Figure 1 of the original paper [5] in terms of how the percent gendered words, percent male bias, and F1 score for each model in each bin compare. Although our results are not exactly the same as those in the original paper [5] in terms of values, the main trends in our results are the same as those in the original paper [5]. The main differences between our results and those in the original paper [5] are lower male bias in each bin for the baseline and a percent gendered words for "CDA" that is closer in value to the baseline in our results.

---

[1]The GitHub repository for our project is located at https://github.com/Pnaghavi/Mitigating-Gender-Bias-in-Generated-Text

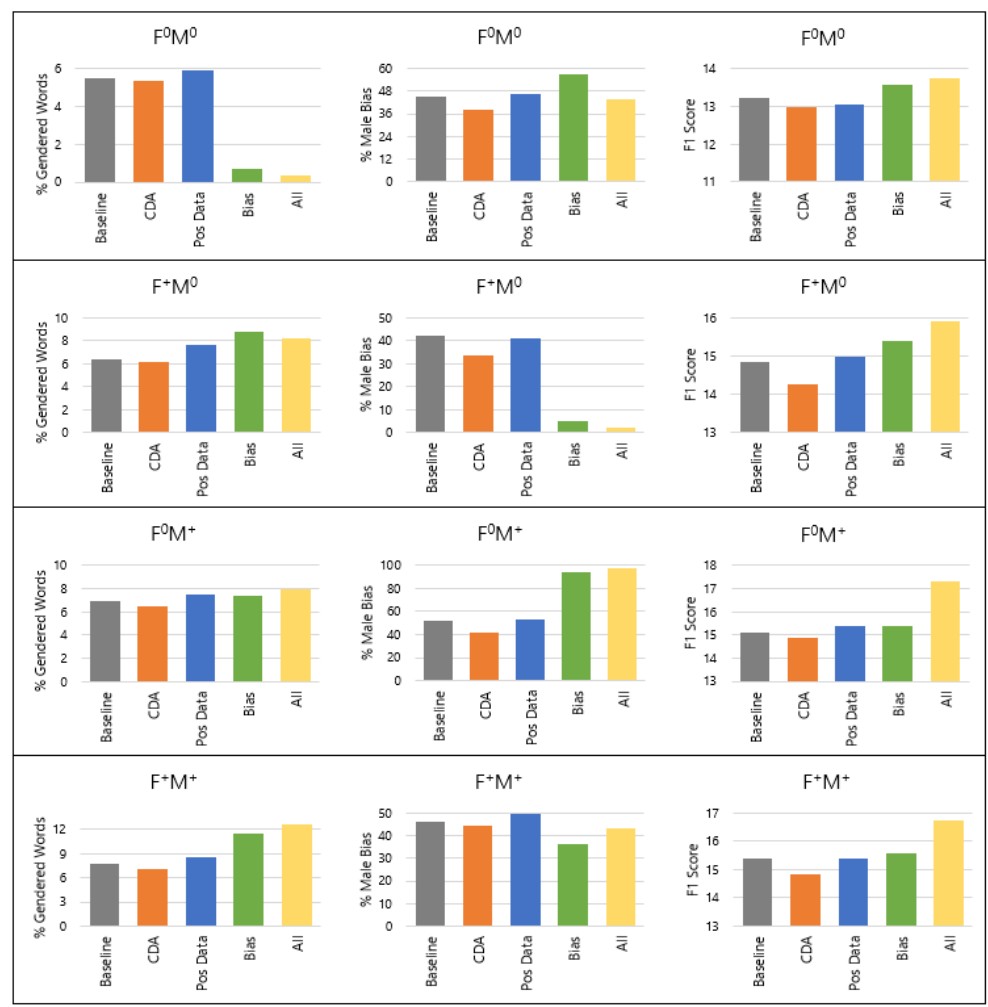

Figure 1: Results for Reproducing the Experiments in the Original Paper [5]

## 4.1 Results for First Hypothesis

According to the first hypothesis, the number of gendered words in the generated utterances for the "All" model for each bin should be similar to the number of gendered words in the labels of the test set. This is observed in all four bins in Figure 1. Specifically, for the $F^0M^0$ bin, the test labels have no gendered words, which means the generated utterances for both models should have a very low number of gendered words and approximately $50\%$ male bias. The "All" model satisfies these two requirements, as depicted in the first set of charts in Figure 1, because the generated utterances from this model are less than $1\%$ gendered words and the percent male bias is approximately $44\%$. For the $F^+M^0$ bin, the test labels have at least one female gendered word and no male gendered words, which means the generated utterances should have a higher number of gendered words and a smaller percentage of male bias. This is observed for the "All" model in the second set of charts in Figure 1, since the percent gendered words for the "All" model is higher than the baseline and the percent male bias is under $5\%$, compared to about $42\%$ male bias for the baseline. Similarly, in the $F^0M^+$ bin, the test labels have at least one male gendered word and no female gendered words. Thus, the generated utterances for the "All" model should have a higher number of gendered words and a larger percentage of male bias, which is depicted in the third set of charts in Figure 1. In the $F^0M^+$ bin, the percent of gendered words for the "All" model is about $1\%$ higher than the baseline and the male bias is approximately $97\%$, compared to only $52\%$ for the baseline. For the last bin, $F^+M^+$, the test labels have at least one male and one female gendered word. As a result, the generated utterances for the "All" model should have a higher percentage of gendered words and closer to $50\%$ male bias. As shown in the last set of charts in Figure 1, the "All" model does have a higher percentage of gendered words than the baseline, specifically $13\%$, compared to $8\%$ for the baseline. However, the male bias is about $43\%$ for

the "All" model, which is not as close to an even gender bias split, $50\%$ male and $50\%$ female, as the baseline, which has about $46\%$ male bias. In the discussion section, we give a possible cause for this discrepancy in our results.

## 4.2 Results for Second Hypothesis

Based on the second hypothesis, the number of gendered words in each utterance generated by the "Bias" model should be similar to that of the labels in the test set for each dialogue. This can be clearly seen for all four bins in Figure 1. In the $F^0M^0$ bin, the test labels have no gendered words. If the model has learned from bias controlled training, producing properly gender biased text according to the bin appended to the end of the dialogue, then the generated text for the "Bias" model in the $F^0M^0$ bin should have very few gendered words and about $50\%$ male bias. As depicted in the first set of charts in Figure 1, for the $F^0M^0$ bin, the "Bias" model has less than $1\%$ gendered words and approximately $57\%$ male bias, as desired. For the $F^+M^0$ bin, the generated text should have more female gendered words and few to no male gendered words, matching the gender bias in the test set label. This is observed in the second set of charts in Figure 1, since the "Bias" model yields a higher percent of gendered words than the baseline and less than $5\%$ male bias, compared to $42\%$ male bias for the baseline. Generated text in the $F^0M^+$ test label bin should have more male gendered words and few to no female gendered words, which is depicted in the third set of charts in Figure 1. Specifically, the percent gendered words for the "Bias" model is $1\%$ higher than the baseline and male bias is approximately $94\%$, compared to only $52\%$ for the baseline. In the last bin, $F^+M^+$, the generated text should ideally have an even distribution of male and female gendered words and a higher percentage of gendered words overall. This is shown in the last set of charts in Figure 1, since the "Bias" model has a higher percentage of gendered words than the baseline, specifically $11\%$ for the "Bias" model and $8\%$ for the baseline, although male bias is $36\%$ for the "Bias" model compared to $46\%$ for the baseline, which is not an even distribution. A possible cause for this discrepancy in our results is described in the discussion section.

## 4.3 Effect of Removing Positively Biased Data Collection

Given the time and monetary cost involved in crowdsourcing data, specifically the positively biased data Dinan et al. collected [5], a natural question is whether adding this positively biased data to counterfactual data augmentation and bias controlled training is worth the cost. In other words, what is the performance loss if positively biased data collection is excluded from the model, instead relying only on counterfactual data augmentation and bias controlled training.

### 4.3.1 Implementation and Experimental Setup

We fine-tuned the model, pre-trained on Reddit conversations [1], on the data generated from counterfactual data augmentation and using bias controlled training. The implementation and experimental setup is the same as that for the model that combines all three bias mitigation techniques, except we excluded the positively biased data collected by Dinan et al. [5].

### 4.3.2 Results and Discussion

Figure 2 depicts, for each bin, the percent gendered words and percent male bias in the generated utterances as well as the F1 score for the "All" model, which combines all three bias mitigation techniques, the "CDA + Bias" model, which uses counterfactual data augmentation and bias controlled training, and the baseline. As expected, for all four bins, the percent gendered words, percent male bias, and F1 score for "All" achieves better results than "CDA + Bias," in terms of higher F1 scores and the percent gendered words and male bias being closer to ground truth, except "CDA + Bias" achieves a slightly higher F1 score for the $F^0M^0$ bin. However, results for "CDA + Bias" are always within about $2\%$ of the results for "All" and the overall F1 score for "CDA + Bias" is within $0.25\%$ of the overall F1 score for "All," specifically an F1 score of $15.31$ for "CDA + Bias" and $15.56$ for "All." Although incorporating positively biased data collection does yield better results, given how small the difference is between including vs. excluding this technique, it may not be worth the necessary time or money. Instead, one could simply use counterfactual data augmentation and bias controlled training or find a less costly way to collect positively biased data, which is the focus of the next extension.

## 4.4 Generating Gender Neutral Data

In the previous section, we created a model incorporating counterfactual data augmentation and bias controlled training, removing positively biased data collection. Instead of completely removing this additional, positively biased data, an alternative, which still avoids the cost of crowdsourcing data, is to generate new, gender neutral data using code. Incorporating gender neutral data can help shift the gender bias of the data, whether male or female, closer to $50\%$.

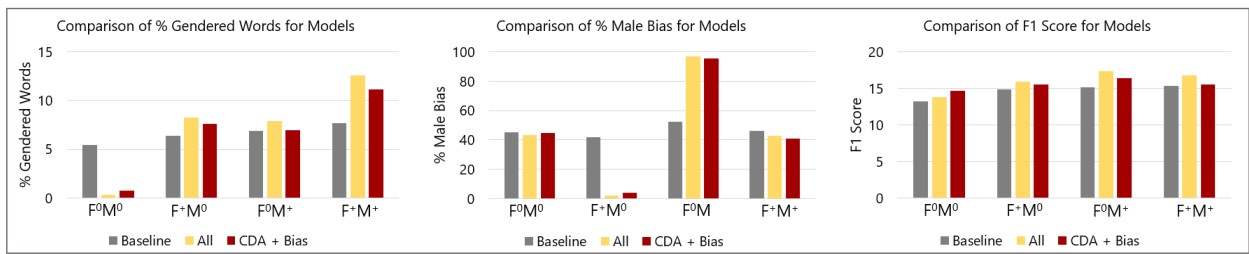

Figure 2: Results for the Baseline vs. Combining all 3 Bias Mitigation Techniques vs. Counterfactual Data Augmentation and Bias Controlled Training

### 4.4.1 Implementation and Experimental Setup

We fine-tuned the model, pre-trained on Reddit conversations [1], using counterfactual data augmentation and bias controlled training, then generated responses from this model for all dialogue episodes in the training data. For each generated response, we set the response to be either the model's generated response or the actual label. If the generated response is neutral, meaning it contains approximately the same number of male and female gendered words or no gendered words, we use the generated response $90\%$ of the time, selecting the actual label in all other cases. These neutral generated responses were used to reconstruct the conversations. We then created new training and validation datasets from these conversations that partially included neutral model generated utterances. Finally, a new model was fine-tuned on these datasets. The experimental setup is the same as that for the model that combines all three bias mitigation techniques, except we excluded the positively biased data collected by Dinan et al. [5] and used the gender neutral data we generated instead. An important point to note is that the test dataset for this new model is the original test dataset. Thus, the F1 scores obtained for each bin and the overall F1 score are from the original test dataset, containing 100% natural conversations.

### 4.4.2 Results and Discussion

Figure 3 shows, for each bin, the percent gendered words and percent male bias in the generated utterances as well as the F1 score for the "All" model, which combines all three bias mitigation techniques, the baseline, and the "CDA + Bias + Our Gen Data" and "CDA + Bias" models, which use counterfactual data augmentation and bias controlled training with and without our neutral, generated data, respectively. Results for our new model, "CDA + Bias + Our Gen Data," are within $2\%$ of the results for "All" in all cases except male bias for $F^0M^0$, $F^+M^0$, and $F^0M^+$. For $F^0M^0$, our model yields male bias closer to $50\%$ than "All" by $6\%$, specifically male bias of about $43\%$ for "All" and $49\%$ for our model. Also, our model results in about $4\%$ higher male bias than "All" for the $F^+M^0$ bin and about $4\%$ lower male bias for the $F^0M^+$ bin. However, these are actually the desired results because for each bin, the male bias for our model is closer to $50\%$, at least slightly, than "All." Thus, our model results in more gender neutral responses overall, which was the goal of this method. In addition, all results for our new model are still relatively close to the results of "All," demonstrating the effectiveness of our new method, as it did not require any crowdsourced data, only additional training. One concern with using model generated responses is that they may not be as coherent as natural dialogue, but the F1 scores for our new model are comparable to those for the "All" model. For future work, if we repeatedly use the dialogues with our neutral, generated responses to create new generated responses, coherency will become a greater concern and necessitate the use of a coherency assessment model, such as some of the machine-learned evaluation metrics highlighted by Celikyilmaz et al. [2]. Given that adding our neutral, generated data to counterfactual data augmentation and bias controlled training yields approximately the same or slightly higher F1 scores than the "All" model, using only neutral, generated responses with high coherency, according to the metrics introduced by Celikyilmaz et al. [2], in the reconstructed conversations, we can continue to shift the model towards gender neutrality, while maintaining high F1 scores.

### 4.5 Percent Generated Responses with Respect to Bins

To better evaluate the degree to which our extensions generate gender neutral responses in comparison to the "All" model, we placed the generated responses from these three models into one of the bias controlled training bins based on the presence of gendered words in the generated response, and computed the percent of generated utterances in each bin for each of the three models.

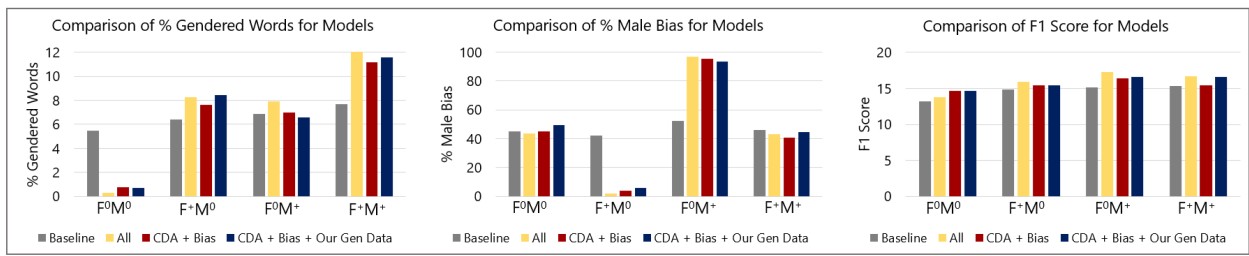

Figure 3: Results for the Baseline vs. Combining all 3 Bias Mitigation Techniques vs. Counterfactual Data Augmentation and Bias Controlled Training both with and without Neutral, Generated Data

### 4.5.1 Results and Discussion

Figure 4 depicts the percent of generated responses in each bin for the baseline, when combining all bias mitigation techniques, denoted "All," and using counterfactual data augmentation and bias controlled training with and without our neutral, generated data, denoted "CDA + Bias + Our Gen Data" and "CDA + Bias," respectively. These results demonstrate that the "CDA + Bias + Our Gen Data" model generates more gender neutral responses overall, compared to "All" and "CDA + Bias." Specifically, for the $F^0M^0$ and $F^+M^+$ bins, which are the more gender neutral bins, "CDA + Bias + Our Gen Data" has the highest, or near highest, percentage of generated responses. For the $F^+M^0$ and $F^0M^+$ bins, which are not gender neutral, "CDA + Bias + Our Gen Data" has the lowest percent of generated responses. In addition to generating more neutral responses, "CDA + Bias + Our Gen Data" achieves approximately the same F1 score for each bin as "All," as depicted in Figure 3, demonstrating that the control over gender bias provided by bias controlled training is still present despite the responses being more gender neutral overall. This indicates an opportunity for future work to shift the overall bias of the model's generated responses to any direction, male biased, female biased, or neutral, by selecting model generated responses that belong to the bin with the desired bias to infuse the original dialogues with this bias and train a model to generate more responses with the desired bias. By repeating this process, we can reinforce the model to generate more responses biased in the desired direction, as long as we can still achieve a high F1 score and maintain coherency, which can be checked by machine-learned coherency metrics [2] as a form of second or outsider opinion on the generated responses during the infusion process.

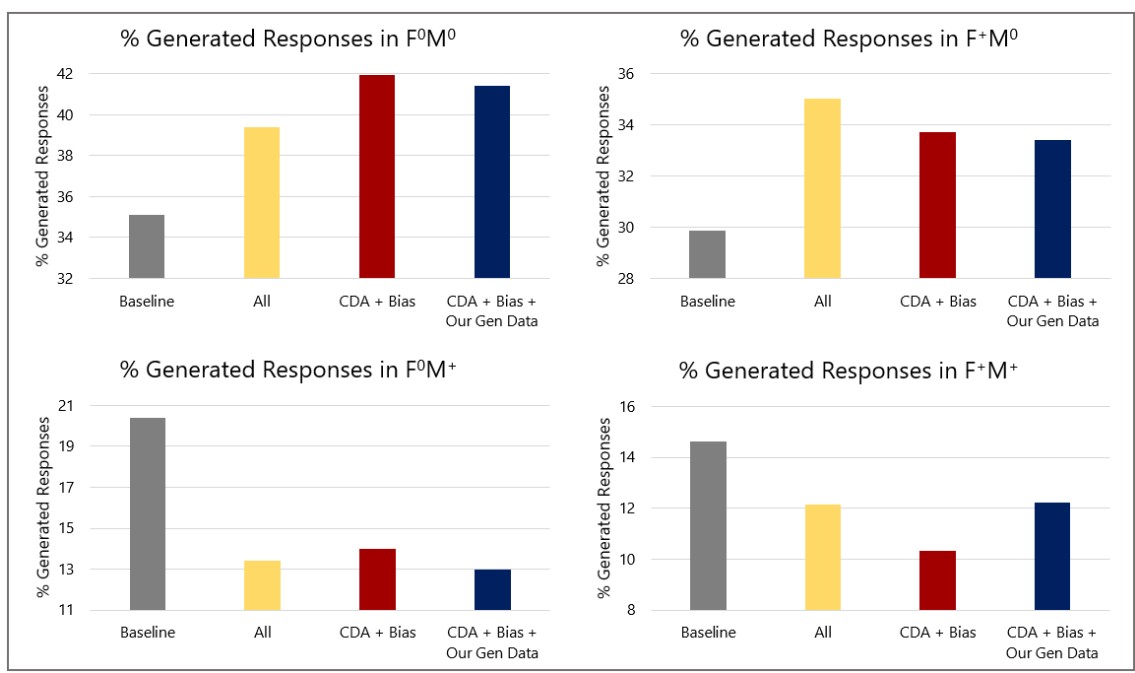

Figure 4: Percent of Generated Responses in each Bin for the Baseline vs. Combining all 3 Bias Mitigation Techniques vs. Counterfactual Data Augmentation and Bias Controlled Training with and without Neutral, Generated Data

# 5 Discussion

Given how closely our experimental results for bias controlled training and combining all three original bias mitigation methods matched the ground truth, these two techniques can be used to control the gender bias of these models' generated text. Thus, gender neutral dialogue could be created by constructing ground truth data with either no gendered words or 50% male bias and 50% female bias within the gendered words. Given that we reproduced the results from the original paper [5] for bias controlled training and combining all three bias mitigation techniques, we feel that overall our results support the claims in the original paper [5], despite the differences in value between our results and those in the original paper [5]. One possible cause for the differences between our results and those in the original paper [5] is our training method, since we achieve higher F1 scores for each model and stop training when perplexity stops decreasing, which may not be the same criteria Dinan et al. used to determine when to stop training. It is also possible that in the original paper [5], the list of gendered words used to place utterances in bins was a subset of the original gendered word list [12], most likely the list of counterfactuals. This could also account for the lower male bias we observed for the baseline in our results compared to Dinan et al.'s, however Dinan et al. explicitly stated they used the gendered word list from Zhao et al. [12]. Evaluating our approach to reproducing the original paper [5], one of the strengths of our approach is that we ran all code on Google Colaboratory with one GPU, a free resource, in a reasonable amount of time. However, Google Colaboratory imposes GPU limitations and as a result, we could not use the same batch size as that in the original paper [5], although we achieve higher F1 scores than those in the original paper [5].

## 5.1 What was easy

When reproducing the original paper [5], implementing counterfactual data augmentation and bias controlled training and combining all three bias mitigation techniques was easy. Specifically, counterfactual data augmentation and bias controlled training were well-described in the original paper [5] and the list of counterfactuals needed for counterfactual data augmentation was provided by Dinan et al. in an easy-to-use format. Combining all three bias mitigation techniques was also an easy part of reproducing the original paper [5], as we simply needed to apply the same techniques used when implementing each bias mitigation method individually.

## 5.2 What was difficult

The only difficulty we encountered, albeit minor, was learning how to use ParlAI, which was necessary in order to use the same model as that in the original paper [5]. However, after reading through the ParlAI documentation and experimenting with the ParlAI Google Colaboratory tutorial [10], we understood how to use ParlAI to fine-tune the model, pre-trained on Reddit conversations [1], for the datasets we created.

## 5.3 Recommendations for reproducibility

Overall, reproducing the original paper [5] was fairly straightforward, but we do have three recommendations to further improve reproducibility. The first is more clearly indicating what model, pre-trained on Reddit conversations, is used, because the source of the model is not provided in the original paper [5], only that the model is based on the implementation by Miller et al. [8], who introduce ParlAI in that paper. The second recommendation is to specify the hyperparameters used when fine-tuning each model, as these were not provided in the original paper [5]. The last recommendation is to describe the stopping condition for fine-tuning the models. We stopped training when perplexity stopped improving, but this resulted in higher F1 scores for the models than those achieved in the original paper [5].

## 5.4 Communication with original authors

We communicated with Emily Dinan, one of the authors of the original paper [5], who clarified what model, pre-trained on Reddit conversations, was used in the original paper [5] and provided us with the command to download the model as well as the hyperparameter settings for training the models.

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

# A   Generated Text Statistics for $F^0M^0$ Bin

| Model | % Gendered Words | % Male Bias | F1 Score | % Generated Responses |
|---|---|---|---|---|
| Baseline | 5.48 | 45.14 | 13.22 | 35.11 |
| Counterfactual Data Augmentation | 5.35 | 38.05 | 12.98 | 38.96 |
| Positively Biased Data Collection | 5.94 | 46.50 | 13.06 | 36.31 |
| Bias Controlled Training | 0.69 | 56.85 | 13.59 | 41.30 |
| All 3 Bias Mitigation Techniques | 0.32 | 43.53 | 13.75 | 39.41 |
| CDA + Bias Control | 0.80 | 44.96 | 14.62 | 41.94 |
| CDA + Bias Control + Our Gen. Data | 0.72 | 49.68 | 14.62 | 41.40 |

Table 2: Results for each Model for $F^0M^0$ Bin

# B   Generated Text Statistics for $F^+M^0$ Bin

| Model | % Gendered Words | % Male Bias | F1 Score | % Generated Responses |
|---|---|---|---|---|
| Baseline | 6.40 | 42.07 | 14.84 | 29.88 |
| Counterfactual Data Augmentation | 6.16 | 33.85 | 14.27 | 31.04 |
| Positively Biased Data Collection | 7.62 | 40.88 | 14.99 | 31.48 |
| Bias Controlled Training | 8.76 | 4.70 | 15.40 | 34.26 |
| All 3 Bias Mitigation Techniques | 8.25 | 1.95 | 15.92 | 35.02 |
| CDA + Bias Control | 7.62 | 4.08 | 15.48 | 33.74 |
| CDA + Bias Control + Our Gen. Data | 8.44 | 5.90 | 15.40 | 33.41 |

Table 3: Results for each Model for $F^+M^0$ Bin

# C   Generated Text Statistics for $F^0M^+$ Bin

| Model | % Gendered Words | % Male Bias | F1 Score | % Generated Responses |
|---|---|---|---|---|
| Baseline | 6.90 | 52.35 | 15.12 | 20.38 |
| Counterfactual Data Augmentation | 6.46 | 41.53 | 14.9 | 18.67 |
| Positively Biased Data Collection | 7.51 | 53.53 | 15.41 | 19.92 |
| Bias Controlled Training | 7.36 | 94.37 | 15.40 | 14.82 |
| All 3 Bias Mitigation Techniques | 7.89 | 97.13 | 17.31 | 13.41 |
| CDA + Bias Control | 6.97 | 95.52 | 16.37 | 14.00 |
| CDA + Bias Control + Our Gen. Data | 6.55 | 93.41 | 16.60 | 12.98 |

Table 4: Results for each Model for $F^0M^+$ Bin

# D Generated Text Statistics for $F^+M^+$ Bin

| Model | % Gendered Words | % Male Bias | F1 Score | % Generated Responses |
|---|---|---|---|---|
| Baseline | 7.70 | 46.28 | 15.38 | 14.64 |
| Counterfactual Data Augmentation | 7.00 | 44.19 | 14.83 | 11.33 |
| Positively Biased Data Collection | 8.51 | 49.71 | 15.37 | 12.28 |
| Bias Controlled Training | 11.40 | 36.41 | 15.56 | 9.62 |
| All 3 Bias Mitigation Techniques | 12.55 | 43.01 | 16.73 | 12.15 |
| CDA + Bias Control | 11.15 | 40.89 | 15.48 | 10.32 |
| CDA + Bias Control + Our Gen. Data | 11.54 | 44.64 | 16.61 | 12.21 |

Table 5: Results for each Model for $F^+M^+$ Bin

# E Distribution of Generated Responses across Bins for each Model

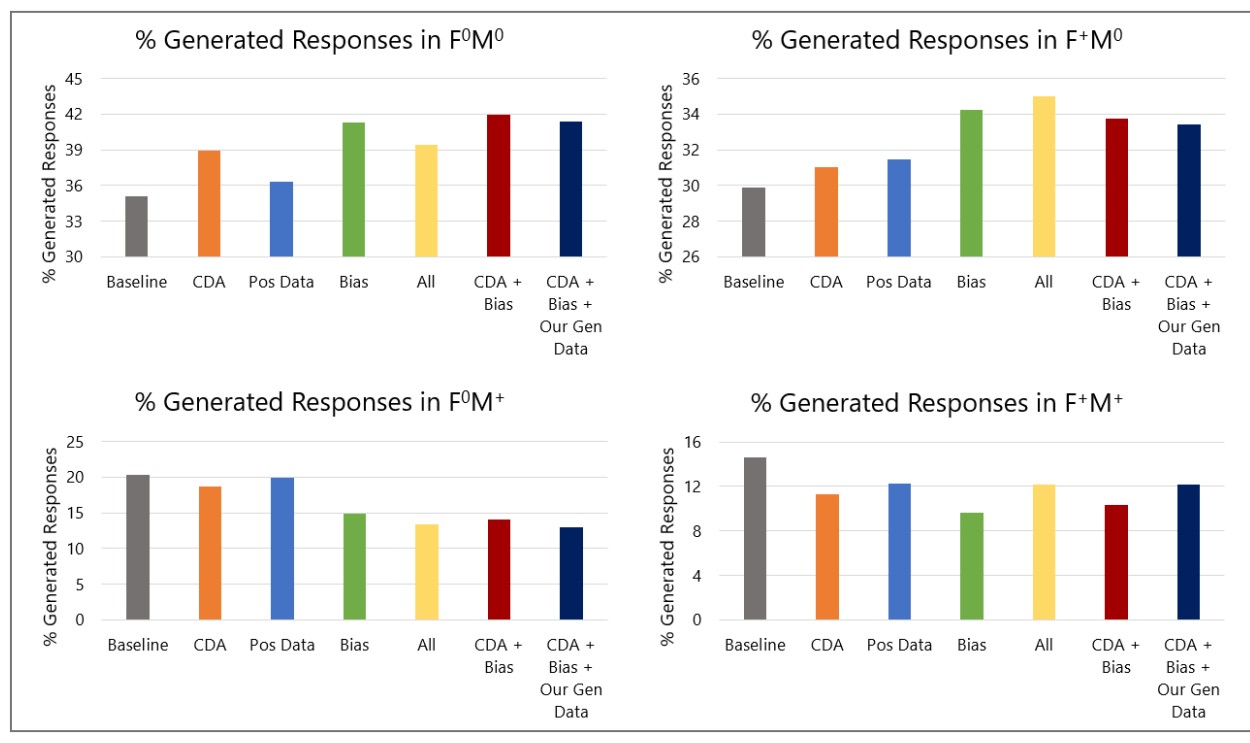

Figure 5: Percent of Generated Responses from each Model in each Bin

