# OpenReview forum: "Reproduction and Extension of "Queens are Powerful too: Mitigating Gender Bias in Dialogue Generation""
_ML_Reproducibility_Challenge/2021/Fall — RC2021_

### Official Review · Reviewer_pBdo · 2022-03-01
**reproduced with better results with ease; originality is low.**

**Rating:** 6
**Confidence:** 4

**Review:**


1. Reproduction results:
The paper does a reasonable job reproducing the results from the original paper [5] for bias controlled training and combining all three bias mitigation techniques. Overall their results support the claims in the original paper.

However, there are the differences in value between their results and those in the original paper. The authors speculate several reasons. (1) There is a difference in training methods. The authors achieve higher F1 scores for each model and stop training when perplexity stops decreasing, while Dinan et al. might be different. (2) It is also possible that the list of gendered words used to place utterances in bins was a subset of the original gendered word list [12]. However, Dinan et al. explicitly stated the contrary. It seems that the authors should be able to dig deeper and get to the root cause of the differences. It is a bit disappointing that this is inconclusive.

2. Reproduction challenges:
It is very surprising there are no major challenges. Kudos to the authors of the original paper!

3. Clarity:
The paper is well written.

4. Quality, originality and significance of the work:
The authors made a good effort and were able to reproduce main results. This is a good contribution. Furthermore, the paper ran all code on Google Colaboratory with one GPU, a free resource, in a reasonable amount of time. Despite Google Colaboratory imposes GPU limitations, the authors achieve higher F1 scores than those in the original paper.

---

### Official Review · Reviewer_TxnS · 2022-03-17
**Well-written and thorough report verifying the findings of the original paper and providing extensions and actionable insights on top**

**Rating:** 9
**Confidence:** 4

**Review:**

The authors of the report attempt to reproduce and extend the results of the paper titled — “Queens are Powerful too: Mitigating Gender Bias in Dialogue Generation”. The original paper first analyzes and examines gender bias in dialogue data (in raw data as well subsequent generative utterances), picks the most susceptible dataset, LIGHT, and then proposes 3 techniques to mitigate gender bias — (1) counterfactual data augmentation, where gendered words are replaced by their corresponding counterparts; (2) positively biased data collection, where (less-biased) new dialogues with female characters and interactions are collected by crowdsourcing; and (3)  bias-controlled training, where the dialog is grouped based on the number of gendered words it contains and the group ID/number is used as a special token during training. This report attempts to re-evaluate two key takeaways from the original paper — (1) do the aforementioned techniques lead to generated dialogues where the proportion of gendered words and male bias closely match ground truth and (2) does bias-controlled training itself control gendered words. Overall, the authors of the report claim that their experiments support the claims made in the original paper. Even though the exact numbers obtained by the authors are not the same, the key trends remain the same as the original paper.

**Strengths**

1. The report is generally very well-written, thorough, and easy to follow. The authors clearly outline the overall summary of the report in the opening “Reproducibility Summary” section. Additionally, the authors do a great job of establishing enough context from the original paper so that it’s straightforward for the reader to follow along for a first read. I particularly appreciate this aspect.
2. As stated earlier, the report is very thorough. I’ll point out a few sections demonstrating this aspect. For instance, in almost every sub-section under the methodology section, the authors are clear and thorough about several design and other choices adopted over the course of their experiments — namely, that they generated training, validation and test splits of the LIGHT dataset, what they pre-trained their model on, specific architectural details of the model, why they used already crowdsourced positively biased data due to time and resource constraints, how certain hyper-parameters were changed due to memory constraints on Google Colab (post discussion with an author of the original paper), how early-stopping was set up based on perplexity and so on. Additionally, the authors also provide the time taken to train individual models (Table 1).
3. The results presented in the report are also quite thorough and help assess the extent to which the claims made in the original paper hold. In addition to the hypothesis outlined in the scope of reproducibility, the authors of the report also study 3 extensions aimed at addressing the high time and monetary cost of crowdsourcing positively biased data. The results presented across several sub-sections demonstrate that the primary claims (that the authors set out to investigate) hold in terms of general trends even though there are slight differences in absolute values (the plots provided, Figures 1-4, are quite useful in this regard). Additionally, the authors of the report attempt to provide some hypotheses behind discrepancies in results (wherever applicable; L188, 206-207, 299-307). In the mini-discussion sections following each sub-result being analyzed, the authors also attempt to provide some actionable insights that might be beneficial for future work in this space (for instance, L285-291).


I don’t have any major weaknesses to point out in the report.

---

### Meta-Review · Area_Chair_rvpF · 2022-04-08

**Recommendation:** Accept
**Confidence:** 5

**Metareview:**

Strong report, good reviews. Although one reviewer was slightly above average, another reviewer gave very convincing points for approval. The discussion of results and insights for improving the reproducibility of the original paper is well formulated and useful for the community. I thus vote for acceptance of the report.

---

### Decision · Program_Chairs · 2022-04-09

**Decision:**

Accept

**Comment:**

Following the recommendation of reviewers and meta-reviewer, the paper is accepted for ML Reproducibility Challenge 2021, and will be published in the upcoming special edition of ReScience Journal.